# Classification of the Acoustics of Loose Gravel [note 1]

**DOI:** 10.3390/s21144944

**Published:** 2021-07-20

**Authors:** Nausheen Saeed, Roger G. Nyberg, Moudud Alam, Mark Dougherty, Diala Jooma, Pascal Rebreyend

**Affiliations:** 1School of Technology and Business Studies, Dalarna University, 78170 Borlänge, Sweden; rny@du.se (R.G.N.); maa@du.se (M.A.); djo@du.se (D.J.); prb@du.se (P.R.); 2School of Information Technology, Halmstad University, 30250 Halmstad, Sweden; mark.dougherty@hh.se

**Keywords:** gravel roads, loose gravel, ensemble bagged trees, sound analysis, road maintenance, GoogLeNet

## Abstract

Road condition evaluation is a critical part of gravel road maintenance. One of the assessed parameters is the amount of loose gravel, as this determines the driving quality and safety. Loose gravel can cause tires to slip and the driver to lose control. An expert assesses the road conditions subjectively by looking at images and notes. This method is labor-intensive and subject to error in judgment; therefore, its reliability is questionable. Road management agencies look for automated and objective measurement systems. In this study, acoustic data on gravel hitting the bottom of a car was used. The connection between the acoustics and the condition of loose gravel on gravel roads was assessed. Traditional supervised learning algorithms and convolution neural network (CNN) were applied, and their performances are compared for the classification of loose gravel acoustics. The advantage of using a pre-trained CNN is that it selects relevant features for training. In addition, pre-trained networks offer the advantage of not requiring days of training or colossal training data. In supervised learning, the accuracy of the ensemble bagged tree algorithm for gravel and non-gravel sound classification was found to be 97.5%, whereas, in the case of deep learning, pre-trained network GoogLeNet accuracy was 97.91% for classifying spectrogram images of the gravel sounds.

## 1. Introduction

In recent years, research on sound recognition systems has gained momentum and has been used in a wide range of applied fields, including health informatics, audio surveillance, illegal deforestation identification, multimedia, animal species identification, and road classification. Audio modality is beneficial not only in the identification of speech and music, recognition of environmental sounds but also in many other areas such as road texture classification can by audio recordings from tires while driving on the roads [1,2,3,4]. Most of the studies for road type detection or road defects detection of paved roads have focused on the paved road [5,6,7]. Some studies have focused on gravel road defects such as dust [8,9]. Only one study was found to have focused on ruts formed by loose gravel [10]. There has been an increase in the deployment of a machine capable of hearing in the environment, such as mobile phones, hearing aids, camera, robots, and wireless microphones. These devices surround us with one or more acoustic sensors/microphones. A microphone is a sensor that converts sound to electrical signals [11]. These sensors can become a part of an acoustic sensor network and can be exploited to solve many speech processing and audio recognition tasks [12].

Through machine learning, systems can make sense of what they hear and help address real-world problems [13,14,15]. Information obtained from the semantic audio analysis can be useful for analyzing, classifying, and predicting an event [16]. Automatic sound analysis systems might not replace experts, but they can support them by processing large data sets and yielding results that facilitate the decision-making process [17].

In Sweden, gravel roads are an economical option for connecting rural populations and providing pathways to recreational locations such as lakes and, most essentially, for facilitating agriculture and forestry activities. Gravel roads make up 21% of all public roads in Sweden, covering 20,200 km. Besides, 74,000 km of existing gravel road and forest roads, out of 210,000 km, are owned by the private sector [18]. On average, these roads have low traffic volume and low average daily traffic (AADT) of below 200 vehicles/day with an average speed of 75 km/h. In contrast, the highest traffic volume in Sweden on a paved road can be found on the E4 Stockholm, with a peak AADT of around 140,000 vehicles/day. The average traffic growth is 2% per year. Gravel road conditions change dynamically; therefore, they need maintenance three to four times a year, usually carried out in the summer when the roads are snow-free [19,20,21].

Evaluations of road conditions are usually subjective and based on photographs taken of the roads and written texts describing the roads. Conditions such as cross fall and road edges are assessed objectively using simple tools like a cross fall meter (2 m long) and a digital meter. Human subjective assessment is not reliable and is prone to error of judgment [22,23,24]. Experts randomly select two 100 m sections over 10 km to decide on the condition of the 10 km gravel road section [25]. This sample size seems very small to decide on conditions for a 10 km road. But due to lack of resources, massive length of gravel roads, and keeping the gravel roads cost-effective, experts do not drive throughout the length of the road, which is under evaluation. The purpose of these evaluations is to plan maintenance activities according to the defect present on the road. These activities could be, blading, treating the road with crushed limestone, removing visibility impairing vegetation, ditching, and salt treatment for dust control. Some expensive objective methods are available such as laser profilers. Laser profilers are trucks with specialized equipment attached to the front and rear of the vehicle, with a cost of up to 500,000 euros; constant availability of such methods is neither economical nor feasible [26]. Smartphone applications are available, but they only tell the overall roughness of the roads, and no information about the defect present. Road maintenance agencies are interested in the defect information to create a customized maintenance plan. Therefore, road maintenance agencies worldwide are looking for cost-effective and efficient objective solutions [27,28].

Internationally, to evaluate gravel roads, quality parameters such as road cross-section, drainage, gradation of gravel, dust, and distress–such as corrugation or wash-boarding, erosion, loose gravel, and potholes–are assessed. One of the crucial parameters that experts look at is loose gravel [29,30]. Loose gravel is a vital assessment factor as too much loose gravel can be dangerous. Loose gravel can slip from under the tires and cause drivers to lose control, resulting in accidents. In 2017, approximately 600 people died of accidents on gravel roads in the United States. Most of these accidents were multiple crashes. One reason for such fatal incidents was excessive loose gravel with other defects such as visibility impairing dust and lack of traffic signs [9]. It also negatively affects both driving quality and comfort. Therefore, it is crucial for road maintenance agencies to have up-to-date information about the condition of gravel roads and plan and execute on-time maintenance of gravel roads.

In addition to their visual assessment capability, humans can get a notion of the amount of loose gravel by hearing gravel hitting the bottom of a car while driving, thus helping them ascertain the condition of the road. We assume that artificial intelligence algorithms can also perform this task. This study investigates whether loose gravel assessment can be improved using acoustic analysis and machine learning algorithms and, if it can, in what way. We hypothesized that the sound recordings of gravel hitting the bottom of the car can provide important information about gravel road conditions. We explore whether collected audio from gravel roads can serve as a predictor for estimating both the amount of loose gravel and the condition of the gravel road by applying pattern recognition techniques. In-car audios of gravel hitting the bottom of the car were recorded while driving on gravel roads. A data set was made by extracting 5 s audio clips. An audio clip where gravel hitting was audible was labeled as gravel sound, and the one with no gravel sounds was labeled as non-gravel sounds. All these recordings were exclusively from driving on gravel roads. For the classification of gravel sounds, sound analysis and machine learning algorithms can be applied to sound recordings from inside the car while it is being driven on gravel roads. Classification results can be visualized on real-time maps to show valuable information about loose gravel conditions on unpaved roads.

We investigated and compared both traditional supervised learning and convolutional neural network (CNN) for the classification of these gravel sounds [31,32,33,34]. In the case of supervised learning, multiple classifiers are evaluated for gravel and non-gravel sound classification. The pre-trained convolutional neural network GoogLeNet was trained and evaluated. Both classifications methods are compared. An ensemble bagged tree-based approach was found to outperform other algorithms in the classification of loose gravel audio signals for gravel road condition evaluation.

The rest of this article is organized as follows: first, we present the theoretical background, which covers studies discussing automated methods, conventional gravel road assessment methods, supervised machine learning algorithms, and convolutional neural networks. Followed by text about data collection and techniques, and then results and discussion. Finally, limitations and future work are discussed.

## 2. Problem Background

The goal of artificial intelligence is to enable computers to make decisions the way that humans can. One of the important human senses is hearing, which allows us to perceive our environment and make decisions accordingly. Humans can readily classify different sounds: music playing, truck engines running, babies crying, people talking, etc. The research area in which machines are used to recognize and interpret sound is called machine hearing [35]. In this study, the utilization of machine hearing is explored so that a different way of assessing loose gravel on gravel roads can be proposed.

Loose gravel can cause tires to slip and the driver to lose control if, for example, they turn the vehicle too quickly. Similarly, drivers need to avoid slamming on the brakes or swerving while driving on gravel roads with lots of loose gravel. Loose aggregate or loose gravel usually results from heavy traffic loads or inferior material, which loosens gravel on the road and the shoulders of the road. Traffic causes gravel to move from the wheel path and form linear berms [29,30]. With loose aggregate, berms can be as big as four to six inches for significant distances along the roadside. Over time, with the accumulation of loose gravel, longitudinal depressions, called ruts or wheel-tracks, form. These ruts can retain water, hindering drainage. The width of the ruts depends on the wheel size, which can vary from a minimum of 6 inches to a maximum of 24 inches [36,37].

Many research studies have been conducted using automated machine learning approaches for identifying road defects. Most of these studies focus on paved roads, while few focus on gravel roads. However, they all focus on detecting the overall roughness of the road and cannot identify the distress type that causes road roughness [38,39]. Moreover, when it comes to distress detection, potholes and rutting distresses are more prominently discussed, and there is no visible work on objective loose gravel measurement [40,41,42,43].

### Diagnostic Standards

Various visual gravel road assessment methods exist worldwide, and these are regulated by the weather and landscape of the country in question. A few prominent methods are the US Army Corps of Engineering Assessment System (USACE), Pavement Condition Index (PCI), and Pavement Surface Evaluation and Rating (PASER) [21,30,36,44]. Municipalities rely on the visual assessment of gravel road conditions and statistical data from vehicles outfitted with specialized instruments that take road surface measurements: examples of such specialized instruments are the laser profiler used in Sweden, the Automated Road Analyzer (ARAN) used in Canada, and the Road Measurement Data Acquisition System (ROMDAS) used in New Zealand [19,21,45]. In Sweden, for gravel road condition assessment, the regulation “Bedömning av grusväglag” (Assessment of Gravel Road Conditions) is used by the Swedish Transport Administration (Trafikverket) [19]. According to this regulation, the evaluation of gravel roads for maintenance purposes is entirely subjective. It aims to acquire information about road conditions and to help decision-makers determine whether or not a gravel road needs maintenance. Photographs of gravel roads are taken from a moving vehicle and later examined by experts: see Figure 1. In addition, written text is included that describes the conditions of certain severity levels for loose gravel: see Table 1 [19].

## 3. Materials and Methods

This section presents the methodology used for this study for data mining and model development. In the summer months, the researchers drove at 50 km/h on gravel roads on the outskirts of the town of Borlänge and in the village of Skenshyttan, both in Sweden, to collect data. Two GoPro HERO7 cameras (GoPro Inc, San Mateo, CA, USA) were used to acquire both sound and visual components. For camera specification details, see Table 2.

One of the cameras was fixed to the windshield inside the vehicle and the other to the vehicle’s bonnet with a double-clip strong suction cup to keep the camera steady without obscuring the camera lens: see Figure 2. The purpose of having two cameras was to see if it was better to record audible gravel sounds inside or outside the vehicle. The data collection involved two trips in dry and sunny conditions. The total distance covered on gravel roads for data collection was 13 miles on the first trip and 10 miles on the second trip. This distance excludes the distance/time to reach the gravel road. Thirty-six minutes of audio recording were found to be useful for the extraction of audible gravel sound data. This number of minutes excludes the time it took to travel to the gravel road. The researchers drove twice in the tracks formed by rutting and twice outside the tracks on the gravel to establish different audibility levels.

Audio data collected using the exterior camera was found to have a lot of wind noise, and no gravel hitting the bottom of the car was audible in the recording. Therefore, for further extraction and analysis of sound, only audio data from the interior camera was used. The video recording from the exterior camera will be used for future studies. The vehicle used for data collection was a Volkswagen Passat GTE (Volkswagen, Wolfsburg, Germany). Details of the vehicle are laid out in Table 3. Figure 3 shows the map of selected gravel roads driven on for sound data collection of in-car gravel sounds.

### 3.1. Supervised Machine Learning Algorithms

Multiple classifiers were evaluated in this study. All the classifiers used in this study are discussed below and are also shown in Figure 4.

#### 3.1.1. Support Vector Machines (SVMs)

A support vector machine is a supervised machine learning algorithm. It is used for classification and regression tasks. The geometric explanation of support vector classification (SVC) is that the algorithm searches for the optimal separating surface, namely the hyperplane, equidistant from the two classes. First, SVC is drawn for the linearly separable case. For non-linear decision surfaces, kernel functions are introduced. A kernel function is a mathematical function that allows SVM to perform a two-dimensional classification of a set of one-dimensional data or else projects data from a low dimensional space to a higher-dimensional space. There are various kernel functions, including linear, polynomial, radial basis, sigmoid, and gaussian. Finally, slack variables are added to allow for training errors when the complete separation of the two variables is not required, for example, for noisy data. SVMs are memory efficient as they use a subset of training points in the decision function, also called support vectors [46,47,48,49]. The SVMs used for this study are linear, quadratic, cubic, fine, medium, and coarse Gaussian.

#### 3.1.2. Decision Trees

Decision tree (DT) learning is one of the approaches used to predict statistics, machine learning, and data mining. A decision tree is used for classification and regression expressed as a recursive partition of the instance space. Classification and Regression Tree (CART) is a term used for both purposes. With classification, data is assigned to a particular class, whereas with regression, the predicted outcome can be considered in numerical form. A decision tree is a method to drive rules from data. Such rule-based techniques are helpful to explain how a model should work to estimate a dependent variable value [50]. A decision tree is a directed tree with a node called a root with no incoming edges. A node with the outgoing edge is called an internal or test node. Nodes other than the internal nodes are called leaves. Each leaf is assigned one class label (decision taken after computing all attributes). Instances are classified by navigating through the root of the tree down to the leaf according to the test results along the path. The paths from the root to leaf represent classification rules [51]. Construction of DT is done by examining a set of training data for which the class labels are already known. DT performs very well on high-quality data. Fine, medium, and coarse trees were used for classification purposes in this study. The maximum number of the splits of these trees is 100, 20, and 4, respectively [52,53,54].

#### 3.1.3. Ensemble Classification

The ensemble-based or multiple classifier techniques are more desirable than the single classifier counterparts as they reduce weak selection possibility [55]. Ensemble methods train multiple classifiers to have a final decision. The ensemble classification method is inspired by the human behavior of considering many experts for a final decision. Many algorithms have been proposed to achieve ensemble learning systems (ELS), such as bagging, boosting, and random forest [56,57]. Ensemble classifiers are categorized into two groups:(i)Classifier selection: The classifier performing best is selected.(ii)Classifier fusion: The output of all the classifiers is combined for the final decision.

Rules are defined so that a class label can be assigned to each instance or, in this case, sound. These rules include weighted majority voting, Borda counting, and behavioral knowledge space common. Ensemble learning has several approaches [58,59,60], such as:(i)Random subspace randomizes the learning algorithm and selects subsets of features from a chosen subspace before training the model. The outputs from the classifiers are combined by majority voting [61,62].(ii)Boosting is a general ensemble method that creates a strong classifier from a number of weak classifiers. A model is built from the training data, and then a second model is created that corrects errors from the first model. Models are added until the training set is predicted perfectly or a maximum number of models are added [63,64].(iii)Bootstrap aggregating is also called bagging. It involves having each model in the ensemble vote with equal weight. To promote model variance, bagging trains each model in the ensemble using a randomly drawn subset of the training set: e.g., the random forest algorithm combines random decision trees with bagging to achieve very high classification accuracy [65,66].(iv)Rotation forest, where every decision tree is trained by applying principal component analysis (PCA) to a random subset of input features [57].(v)We used ensemble bagged tree (EBT) based classification in this study. Bagging is considered highly accurate and the most efficient of ensemble approaches. Bagged decision trees can improve the performance of decision trees since they aggregate the results of multiple decision trees. In a given data set, bootstrapped subsamples are drawn, and a decision tree is established on each bootstrapped sample. The result of each decision tree is aggregated to yield a robust and accurate predictor [67,68].

### 3.2. Data Preparation and Pre-Selection of Gravel and Non-Gravel Sound Events

First, the audio signal was extracted from the mp4 video files and stored in .wav format, which had a 44,100 Hz sampling frequency with 16-bit per sample. We applied audio pre-segmentation, which is generally the task of separating a continuous audio stream into small audio portions, also called segments. The length of each segment was set to 5 s to provide the necessary information to perform the experiments. In the next step, all the audios were sorted into two classes labeled as gravel or non-gravel. We describe these audio groups as follows:

#### 3.2.1. Gravel Sound

Gravel sound is an audio recording in which gravel hitting the bottom of the car is audible. These audio clips are obtained from gravel roads with conditions shown in images 1 and 2 of Table 1.

#### 3.2.2. Non-Gravel Sound

Non-gravel sound is an audio recording in which gravel hitting the bottom of the car is either not audible or is audible once or twice. These recordings are primarily obtained more from road-types 3 and 4 of Table 1. Therefore, we can say that these audio clips represent the same roads.

Both gravel and non-gravel sounds were from audio recordings from gravel roads exclusively. Audio clips disturbed by non-static background noise, such as speech, environmental sounds, the sound of the car indicator, or the sound of the horn, were excluded. A sound recording was discarded whenever a gravel or non-gravel acoustic could not be extracted from the original recording. Both of the classes consist of audio recordings from gravel roads only. In this study, 237 audio clips were used, comprising 133 gravel sounds (56%) and 104 non-gravel sounds (44%).

Variants of supervised learning algorithms (SVMs, trees, ensemble algorithms) were trained for classification tasks. Feature extraction was done in the R statistical computing language [69]. All classification and visualization tasks were performed in the Matlab classification learner app, an interactive app, and provides an opportunity to test several classifiers with a graphical user interface (GUI). Because of its interactive GUI, Matlab was used to train and test. Feature extraction, training, and testing can also be completely done in R.

### 3.3. Signal Processing and Feature Selection

The audio data is by default in the time domain. It was converted to the frequency domain by Fast Fourier Transform (FFT) to look for patterns [70]. Computing the Fast Fourier Transform (FFT) on the whole sound or a single section might not be informative enough. An intuitive solution is a Short-Time Fourier Transform (STFT) that computes the Discrete Fourier Transform (DFT) on subsequent sections along with the signal.

A window is then sided along with the signal, and a DFT is calculated at each slide or jump [32]. STFT was performed on audio data in this study. One of the drawbacks of performing STFT is that it introduces artifacts such as side frequency lobes at the edges while sliding through the signal. A windowing function called the Hamming window was used for our data with a 50% overlap to avoid any spectral leakages. The Hamming window is a taper formed by raised cosine with non-zero endpoints optimized to minimize the nearest side lobe, which provides a more accurate idea of the original signal’s frequency spectrum.

Figure 5 below shows signals from both sound classes, i.e., gravel and non-gravel, in the frequency domain. Most of the audio information exists within the 2 kHz range as other non-speech audio. Amplitude differences can be seen between the two sample sounds. With audio of gravel, more frequencies with higher amplitudes can be observed.

We extracted 79 spectral/frequency domain features, such as spectral centroid, amplitude, harmonics-to-noise ratio, mean frequency, and peak frequency (mean, median, and standard deviation).

A *t*-test was performed for dimensionality reduction of the features and for the selection of features that show a significant difference between the classes. Some of the features, having a significance value below the *p*-value 0.05, were selected for model training purposes. A response variable class (gravel, non-gravel) and 36 predictors were picked with a *p*-value below 0.05. Features with a *p*-value of greater than 0.05 were not selected in the classification process.

Principal component analysis (PCA) was also tested for dimensionality reduction. With PCA, nine features were selected. These nine features achieved a classification accuracy of 56% to 58%. Hence, feature selection by the *t*-test was finally chosen.

Ten-fold cross-validation was used to assess the accuracy of each model. This method partitions the data into ten subsets while maintaining the proportionality of each class. Nine subsets were used to train the models, and the tenth subset was used to test accuracy. This method was repeated until all the subsets were used as training and test sets.

### 3.4. Classification of Audio Spectrograms Using Convolutional Neural Networks (CNN)

In addition to investigating traditional supervised learning methods, we will also investigate loose gravel sound classification using Convolutional Neural Network (CNN). This investigation will compare both methods.

Gravel acoustics were converted to spectrogram images by Fast Fourier Transform (FFT). A pre-trained network GoogLeNet was trained using these images with some fine-tuning to the network. GoogLeNet is a 22-layer deep convolutional neural network. It is a variant of the Inception Network, a deep convolutional neural network developed by researchers at Google [31,33]. We will discuss GoogLeNet later in this section. The details of the technologies are discussed in this section.

#### Spectrograms

Sound waves are made up of high and low-pressure regions moving through a medium. Such pressure patterns make the sound distinguishable. These waves have characteristics such as wavelength, frequency, speed, and time. Machines can classify sounds based on such characteristics, just as humans do [70,71].

A spectrogram is a way to visualize a sound wave frequency spectrum when it varies over time. We can say it is a photograph of the frequency spectrum that shows intensities by varying colors or brightness. One way to create a spectrogram is through the use of FFT, a digital process. We have used this method to generate spectrograms in this study. Digitally sampled data in the time domain is broken into segments, usually using overlap and Fourier transformed data to calculate the magnitude of the frequency spectrum for each chunk. Each chunk corresponds to a vertical line in the spectrogram. These spectrums are laid side by side to form the image or three-dimensional surface with information of the time, frequency, and amplitude [72]. Amplitude is shown by using intensities of colors; brighter colors show higher frequencies of sound waves. Spectrograms of gravel and non-gravel sound are shown in Figure 6.

Neural networks (NN) are inspired by the human brain. A neural network comprises many artificial neurons containing weights and biases. These networks learn feature presentation, thus eliminating the process of manual feature selection process [63]. The training process involves backpropagation to minimize a loss of function,  L=g(x,y,θ)  through the tuning of parameters, θ. A loss function is calculated as the difference between observed and actual values. The cross-entropy loss function is often a choice in classification problems. The loss function is optimized iteratively through the calculation of the gradient descent by learning rate. The learning rate is an important parameter; it is the rate at which the gradients of each neuron are updated. A higher learning rate can reach the goal quickly but risks reaching a local minima [73,74,75,76,77]. The goal of the loss function is to reach a global minimum acceptable value for the loss function. The most common optimizers are stochastic gradient descent and its variants. These networks are composed of connected layers, each layer having many neurons. Deep neural networks (DNNs) are referred to as NNs with many layers. Multiple layers enable them to solve complex problems that their relatively shallow networks usually cannot solve. The network depth seems to contribute to the improved classification [78,79].

In several studies, CNNs classify spectrograms for musical onset detection, classification of acoustic scenes and events, emotion recognition, or identification of dangerous situations in underground car parking to activate an automatic alert from sound [80,81,82,83,84]. Convolutional neural networks (CNNs) have become popular in machine learning research. CNN’s are widely applied to visual recognition and audio analysis. CNN’s consist of specialized layers for feature extraction images called convolutional layers. Convolutional layers have filters to learn features such as edges, circles, or textures. Each convolutional layer convolves the input and passes the result to the next layer, resulting in a complex feature map of the image [85].

One of the first CNNs was LeNet. It was used to recognize digits and characters. LeNet architecture includes two convolutional layers and two fully connected layers [86]. One reason for the success of CNNs is their ability to capture spatially local and hierarchical features from images. Later, a deeper CNN was proposed called AlexNet, which achieved record-breaking accuracy on the Imagenet large-scale visual recognition challenge (ILSVRC-2010) classification task [87]. In addition to having increased depth, AlexNet also has a rectified linear unit (ReLU) as its activation function and overlapping max pooling to downsample the features of the layers.

Training CNNs requires a considerable amount of data and time, which in most cases are not available. Using a pre-trained network with transfer learning is typically much faster and easier than training a network from scratch. Pertained networks are CNNs with descriptors that are extracted by training on large sets. These descriptors from pre-trained networks can help in many visual recognition problems with high accuracy [88].

Many pre-trained networks are developed over time, such as a residual neural network (ResNet), AlexNet, GoogLeNet, FractalNet, VGG, etc. These networks are trained on different data sets and have variants depending on the number of layers in the architecture. Pre-trained networks are trained on millions of images from data sets that are publicly available. The training requires a considerable amount of computational power and may take weeks of training depending on the network architecture’s complexity. By taking advantage of transfer learning from pre-trained networks, other classification problems can often be solved by fine-tuning pre-trained networks. Fine-tuning is the task of training and tweaking a pre-trained network with a small data set and fewer classes than the pre-trained network [89].

For this study, the dataset is considerably small (i.e., 237 spectrograms images) for training a network from scratch. We can still take advantage of pre-trained convolutional networks. Data augmentation is a technique used to artificially create new training data from existing training data [90]. We also used data augmentation techniques, such as image resize, horizontal flip, and random rotation. We increased the image data set four times with data augmentation and fed it to the CNN as four different sets of images. Each image’s dimensions were 224 × 224 pixels, as it is the default input images size required by GoogLeNet.

We used GoogLeNet for the classification of spectrograms of gravel acoustics. GoogLeNet or Inceptionv1 was proposed by Google research in collaboration with various universities. GoogLeNet architecture outperformed its counterpart in classification and detection challenges in the ImageNet Large-Scale Visual Recognition Challenge 2014 (ILSVRC14). It provided a lower error rate than AlexNet, the previous winner of the challenge in 2012. GoogLeNet architecture consists of 22 layers. It introduced various features such as 1 × 1 convolution and global average pooling that reduce the number of parameters and create a deeper architecture. GoogLeNet is a pre-trained network trained on the ImageNet dataset, comprising over 100,000 images across 1000 classes. The large data set of ImageNet contains abundant examples of a variety of images. Feature knowledge gained by GoogLeNet could be practical in the classification of the images of other data sets. In this study, we leverage this knowledge of GoogLeNet gained from training on larger data sets of images to help classify spectrograms of gravel audios with a relatively small data set of 237 audio spectrograms. This method can help in achieving better results. More details about GoogLeNet architecture can be found in the paper in the following reference [91].

## 4. Results

We used both classical machine learning algorithms and convolutional neural networks (CNN) for classification purposes. In this section, we discuss the results from both methods.

### 4.1. Results from Supervised Learning

The extracted audio recording consists of the sound of gravel hitting the bottom of the vehicle. Data sets of extracted audios were labeled gravel and non-gravel sounds. Features were extracted and saved into a .csv file and were analyzed by *t*-test. Features that had significant differences between the classes were selected for training. Selected features were fed for training and classifications to the classifiers.

Classification results by different classifiers are presented in Table 4, which shows the accuracy of the algorithms used. Accuracy can easily be defined as $\mathrm{Accuracy}=\frac{\mathrm{TP}\;+\;\mathrm{TN}}{\mathrm{TP}\;+\;\mathrm{FP}\;+\;\mathrm{FN}\;+\;\mathrm{TN}}$, where TP is the true positive rate, and TN is the true negative rate (TN = 1 − FP) [80]. Ensemble bagged trees (EBT) outperform all other algorithms with 97% accuracy. Accuracy is the number of correct predictions made divided by the total number of predictions made, multiplied by 100 to turn it into a percentage. EBT also performs better than others in classifying both positive and negative classes, i.e., it has an accuracy of 99% in classifying gravel class and 94% for non-gravel sounds, as shown in Figure 7. In EBT, the misclassification of non-gravel is almost five times greater than that of gravel audio, but still, the classification rate is incredible. This misclassification could result from some non-gravel audio having few gravel-hitting sounds that could be classified as gravel sounds.

Quadratic SVM performed the best in the SVM group, and fine trees performed with almost the same accuracy. Figure 7 shows a comparison of the performance of algorithms and presents the true positive detection rate of all the classifiers/algorithms for both gravel and non-gravel. A true positive is an outcome where the model correctly predicts the positive class. Similarly, a false positive is an outcome where the model incorrectly predicts the positive class.

### 4.2. Results from CNN

The use of the convolution layers helps avoid the feature extraction process necessary for classical ML algorithms. However, these still require a lot of data to avoid over-fitting. When the training data is scarce, alternative methods are needed. Transfer learning is a machine learning technique that transfers knowledge learned from a source domain to a target domain [77]. It is an advantageous method to avoid over-fitting when the task-related data is small. There have been successful attempts in the literature to apply transfer learning in classification tasks [89]. We also employ transfer learning to train our CNN. Spectrograms of audio data were generated for both classes of gravel and non-gravel sounds. Data augmentation techniques such as random horizontal flip, random rotation, and resize were applied to the data set. After each augmentation technique is applied, a new set of images is created to be submitted to CNN [92]. It improved the performance in terms of accuracy of the pre-trained CNN, in this case, GoogLeNet. The accuracy achieved was 97.91% using GoogLeNet pre-trained network, as visualized in Figure 8. Changes were made to the last fully connected layer of the pre-trained architecture to classify only two classes in our case, i.e., the gravel and non-gravel. The network was trained for 100 epochs with a learning rate of 1 × 10^−3^. Decreasing the learning rate also improved the accuracy. Figure 8 shows that the architecture showed some overfitting at the beginning of the training process, but after 20 epochs, the accuracy became stable. Using convolutional networks also shows good accuracy as supervised learning algorithms. The benefit of using CNN for classification in this case and in general is that the neural network is responsible for extracting and selecting appropriate features for training the CNN for classification, making the process simpler and more efficient.

In terms of computational cost, classical methods were trained on core i5, and it took around 40 s for the training and testing of each algorithm. In addition, the feature extraction process took 30 min. A better machine would cut this time significantly. GoogLeNet was trained on Google Colaboratory with graphical user interface (GPU) Nvidia K80/T4 to avoid hours of training on the same machine. It took around 30 min to get training and validation results. Google Colaboratory is a cloud disseminating machine learning and research service. It lets the users use GPU/TPU through cloud services. The run time of Google lab is configured with artificial intelligence (AI) libraries. The service is linked to google drive [93]. CNN is more computationally expensive than EBT or other algorithms. On the other hand, the benefit of using CNN over the other algorithms used in the study would be that it takes care of feature extraction and feature selection processes. Any of the methods of EBT or CNN can be used depending on the availability of computational resources.

## 5. Conclusions, Limitations, and Future Work

A great deal of literature includes studies on road condition monitoring systems and primarily focuses on paved roads. Gravel road condition assessments also need to be considered. In automated systems, road condition distresses are mainly identified by images or accelerometer data. However, applications using acoustic data have not been comprehensively explored. This study shows that an objective assessment of gravel roads for loose gravel through acoustic data is promising. Acoustic signals collected by driving on gravel roads render valuable information about road conditions when the loose gravel parameter is considered. Moreover, this can be achieved by cost-efficient methods involving acoustic sensor/mic of simple equipment such as a camera or other portable recording device.

Applications with such machine models can classify gravel sounds. These classification results can be visualized on gravel roads on maps to show the extent of loose gravel along the gravel roads. Citizens can also use such applications to share real-time data and plan their trips. They can know the road conditions in advance. People frequently using gravel roads may be interested in such applications and can provide data of longer length by just using the app on their drive. These applications can also help maintenance agencies have real-time data and plan for timely and defect-specific road maintenance plans.

Both supervised learning and CNN were used, and results were compared for this study. In classical algorithms, ensemble bagged tree-based classifiers perform best for classifying gravel and non-gravel sounds among various classifiers. EBT performance is good in reducing the misclassification of non-gravel sounds. The use of CNNs also showed 97.91%. Using CNN makes the classification process more intuitive as the network architecture takes the responsibility of selecting relevant training features. The classification results can be visualized on road maps, which can help road monitoring agencies assess road conditions and plan road maintenance activities for a particular road.

The gravel sounds were easily detected when the vehicle was driven outside the tracks or ruts on the loose gravel. For accurate recordings of gravel audio, driving outside the tracks during data collection is to be recommended.

In this study, one limitation is that all the recordings were made using one vehicle, a Volkswagen Passat GTE. Other vehicles might produce more or less engine sound, and as such, the results might vary. Therefore, the addition of more recordings from different vehicles will most likely provide deeper insight into how the proposed system detects gravel sounds for audio recording from other vehicles. As a continuation of this work, research will be conducted using machine vision to classify loose gravel by applying deep neural networks. A fusion of video and sound data could be researched for better results. This method can be used on gravel roads in countries with similar terrain and weather conditions as Sweden.

## Figures and Tables

**Figure 1 sensors-21-04944-f001:**
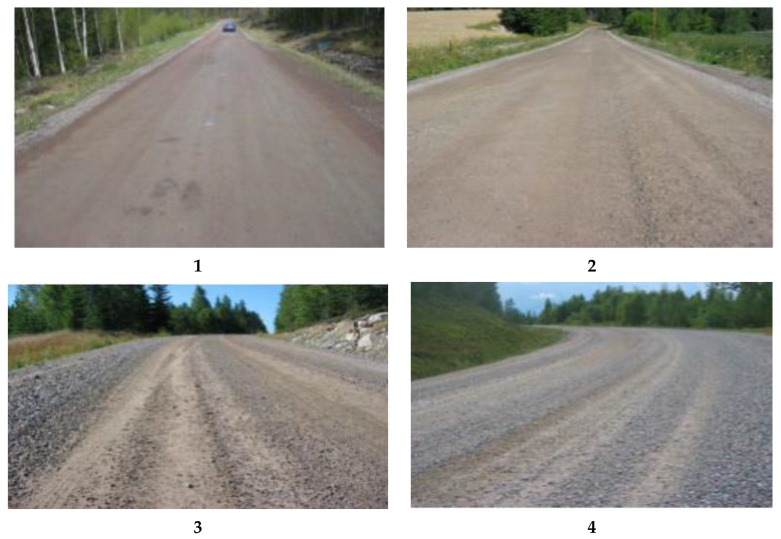
Example images of gravel roads for the Swedish gravel road assessment manual defined by Trafikverket showing the severity level for loose gravel: from **1** “good condition” roads to **4** “worst condition” roads. These images provide a guideline/ground truth for the experts to compare real-time images with these standard images and assign a rating to a particular road under observation. Maintenance plans are decided according to these ratings for each road. Detailed descriptions of each road type are defined in Table 1 [19].

**Figure 2 sensors-21-04944-f002:**
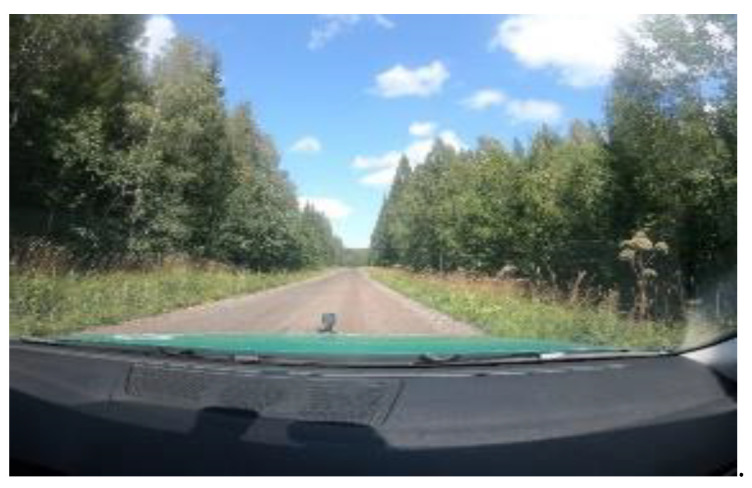
This photograph was taken using the camera inside the car and shows the camera on the car’s bonnet. The camera installed inside was intended to record in-car audio recordings of gravel hitting the bottom. While the camera outside had better video recordings of the gravel road.

**Figure 3 sensors-21-04944-f003:**
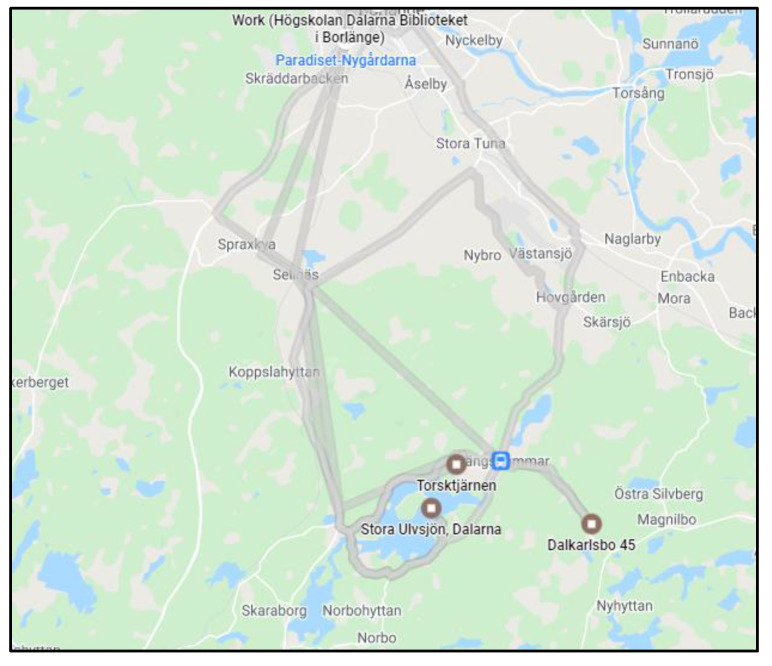
Map showing the trip where data collection took place on gravel roads on the outskirts of Borlänge and in Skenshyttan, Sweden.

**Figure 4 sensors-21-04944-f004:**
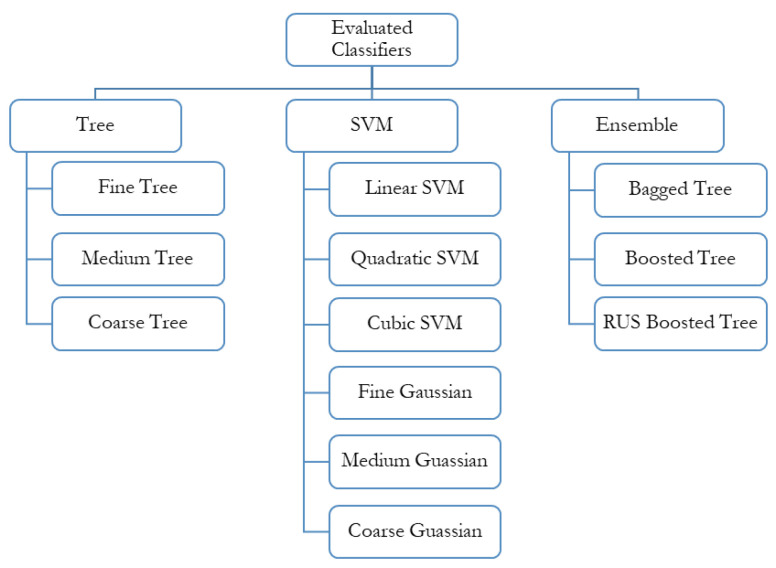
Algorithms used for classification by supervised learning.

**Figure 5 sensors-21-04944-f005:**
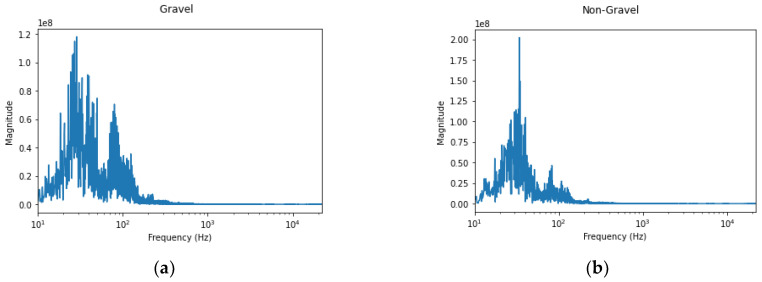
Example of processing of the recorded signals. Sound signal of gravel (**a**) and non-gravel (**b**) in the frequency domain. More high magnitude frequencies are observed in gravel sounds.

**Figure 6 sensors-21-04944-f006:**
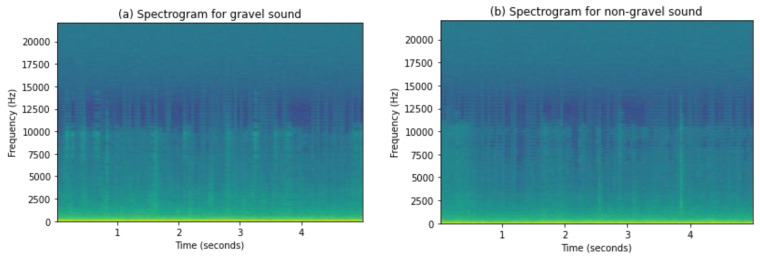
Spectrogram images of non-gravel and gravel sound.

**Figure 7 sensors-21-04944-f007:**
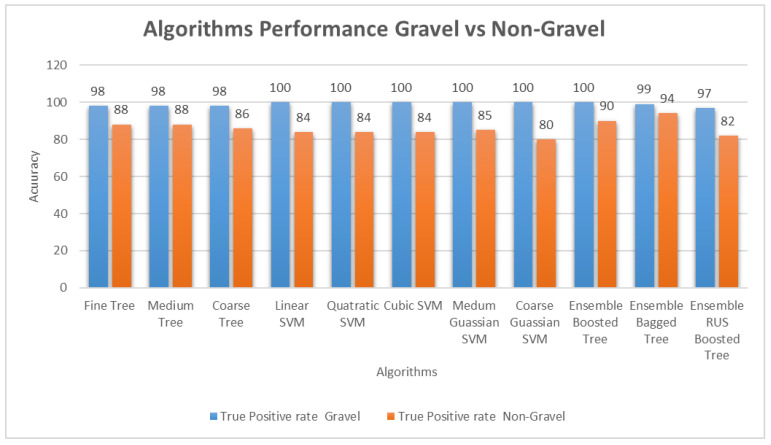
Classification performance of the classical algorithm. The figure shows the true positive rate of detection of both classes. EBT outperforms all the other algorithms in classifying instances of both classes.

**Figure 8 sensors-21-04944-f008:**
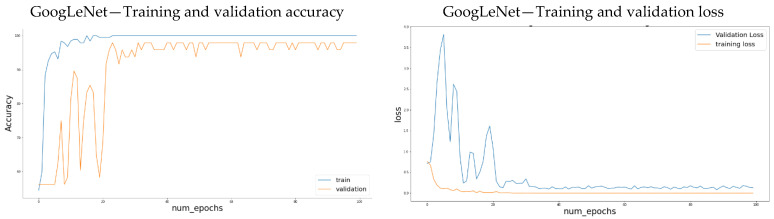
Training and validation accuracy and loss plots obtained from CNN.

**Table 1 sensors-21-04944-t001:** In addition to images, some text descriptions are laid in the manual for the Swedish gravel road assessment method explaining each severity level to aid experts in deciding on a certain road condition [19].

Severity Level	Description
1	No loose gravel on the road, but there may be a small amount along the roadside.
2	A small amount of loose gravel on the road and in small embankments along the roadside, but this does not affect driving comfort or safety to any notable degree.
3	Loose gravel on the road and in small embankments along the roadside significantly affects driving comfort and safety.
4	An extensive amount of loose gravel on the road and in marked embankments at the edge of the road affecting driving comfort and safety.

**Table 2 sensors-21-04944-t002:** Audio and video specification of GoPro Hero 7 Black camera(s) used during data collection.

GoPro Hero 7 Black Specifications Used	
Video	Audio
Supported shooting formats 4K/60 fps, 4K(4:3)/30 fps, 2.7k/120 fps, 2.7K(4:3)/60 fps, 1440p/120 fps, 1080 p/240 fps, 960/240 fps, 720/240 fps Shooting angles 150°	Bit rate 164 Kbps Channels 2 (stereo) Audio sample rate 48 kHz

**Table 3 sensors-21-04944-t003:** Specification details of the vehicle used during data collection.

Specification of the Vehicle Used during Data Collection	
Manufacturer	Volkswagen
Model	Passat GTE 2018
Weight	1806 kg
Power	115 kW
Engine type	Plugin hybrid engine
Engine size	1395 cm^3^/1.4 L
Tire type	Summer tires
Tire dimensions	215/55 R17 94V
Ground clearance	14.5 cm/5.71 inches
Sound level	70 dB still and 73 dB while driving
Maximum speed	225 km/h

**Table 4 sensors-21-04944-t004:** Accuracy of various algorithms used in this study for the classification of gravel and non-gravel sounds.

Model	Accuracy (%)
**Decision Trees**	
Fine Tree	93.2
Medium Tree	93.2
Coarse Tree	92.4
**Support Vector Machine**	
Linear SVM	92.8
Quadratic SVM	93.7
Cubic SVM	92.8
Medium Gaussian SVM	93.2
Coarse Gaussian SVM	91.1
**Ensemble Classification**	
Boosted Tree	95
Bagged Tree	97
RUSBoosted Tree	90.3
Convolutional neural network (CNN)	97.91

## Data Availability

The data used to support the findings of this study are available from the corresponding author upon request.

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
