# Peer review of "Classification of the Acoustics of Loose Gravelâ€"

_sensors, 2021, doi:10.3390/s21144944_

Round 1
Reviewer 1 Report
In this study, acoustic data on gravel hitting the bottom of a car was used. The connection between the acoustics and the condition of loose gravel on gravel roads was assessed. Traditional supervised learning algorithms and convolution neural network (CNN) were applied, and their performances are compared for the classification of loose gravel acoustics.
Section 1 must be improved. Introduce better the topic of sound detection represents the field in which your work is positioned so you should dwell more. Also introduce acoustic sensors, it's okay to talk about devices such as mobile phones and cameras, but in reality, what interests us is precisely the acoustic sensor. Then move on to effectively introduce the problem you are facing, first explain to readers why gravel can be a problem and then describe how the problem could be addressed. Explain how sound detection can help us. Finally, add references to works that have also tackled the problem from other points of view.
Section 2 must be improved. Describe in detail the equipment used to make the measurements. Extract this data from the datasheet of the instrumentation manufacturer. To make reading the specifications of the instruments more immediate, you can insert them in a table, listing the instruments used and the specific characteristics for each. Describe in more detail the procedure followed for data collection (length traveled, how long, etc.)
Section 3 must be improved.
You need to be clearer in labeling data. This is the essential part of the job, the labeling. Explain what you mean by gravel or non-gravel. Did you just operate this simple dichotomous labeling? What do you mean by non-gravel? Maybe an asphalted road? In this case, the whole identification procedure is quite trivial, it is clear that traveling on a dirt road is noisier than traveling on an asphalted road. It would have been more interesting to identify the state of the dirt road, I mean the amount of gravel present on the track and the diameter of the cobbles.
Section 4 must be improved. You should enrich the information on the results obtained. You should properly introduce the evaluation metrics used. In order to make an effective comparison you should also enter information on the computational cost in order to understand if it is appropriate to use more expensive technologies from a computational point of view.
Section 5 must be improved. Paragraphs are missing where the possible practical applications of the results of this study are reported. What these results can serve the people, it is necessary to insert possible uses of this study that justify their publication.
31-43) Introduce better the topic of sound detection represents the field in which your work is positioned so you should dwell more. Also introduce acoustic sensors, it's okay to talk about devices such as mobile phones and cameras, but in reality what interests us is precisely the acoustic sensor.
31-34) Add references to allow readers to learn more about these topics.
44-47) Then move on to effectively introduce the problem you are facing, first explain to readers why gravel can be a problem and then describe how the problem could be addressed. Explain how sound detection can help us. Finally, add references to works that have also tackled the problem from other points of view. Move this paragraph after line 76
48-50)Move this paragraph after line 76
77-84) Briefly explain how the signs have been labeled.
85-91) Add references to allow readers to learn more about these topics.
122) Add subsection number
132) "Bedömning av grusväglag" Add reference
140-141)Where are 2 and 3 labels? Add a reference to Table 1 connection in the caption.
147) Add a table with GoPro HERO7 cameras specifications (Audio and Video)
161-164) So the camera on the hood was used only for video, while the internal one for audio. Specify it better.
164) Add a table with Volkswagen Passat GTE specifications: ground clearance, type of tires, operating speed, etc. These are all characteristics that determine the type of noise produced inside the car and therefore for the reproducibility of the experiment they must be specified in detail.
168) Figure 3 is too genric to try to focus the image on the stretch that the researchers have traveled. Specifies the length traveled and for how long.
176) Do not use abbreviation such as i.e.
178) kernel functions. Introduce adequately the topic
180) Do not use abbreviation such as e.g.
185-193) Introduce adequately the topic, add more reference, You must enter a detailed bibliography which studies you refer to.
208-221) Add references to allow readers to learn more about these topics.
234) “as gravel or non-gravel”. This is the essential part of the job, the labeling. Explain what you mean by gravel or non-gravel. Did you just operate this simple dichotomous labeling? What do you mean by non-gravel? Maybe an asphalted road? In this case, the whole identification procedure is quite trivial, it is clear that traveling on a dirt road is noisier than traveling on an asphalted road. It would have been more interesting to identify the state of the dirt road, I mean the amount of gravel present on the track and the diameter of the cobbles.
234-236) If the classification concerns only these two classes you could also leave us the noise.
237-238) Explain better what you mean
238-239) So the classification was done by having a human hear the recordings made on the road?
250-251) Why Feature extraction in R and classification in Matlab?
292) Convolutional Neural Network (CNN). Introduce adequately the topic, do not forget the work of Yann Le Cun from which it all began. I saw the 57 reference
318-330) Add references to allow readers to learn more about the topic
334-335)Add more reference that used CNN.
410) Why didn't you include CNNs in this results table as well?
414-435) In order to make an effective comparison you should also enter information on the computational cost in order to understand if it is appropriate to use more expensive technologies from a computational point of view.
Reviewer 2 Report
- The way of reference marking is unreasonable. In line 38, the references are marked as [1], [2], [3], [4], [5]. However, in line 62, the references are marked as [8]-[12]. I am wondering which way fits the requirements of the journal.
- Are there other scholars using machine learning methods to achieve loose gravel assessment? The authors should present more references about this research field in the Introduction, especially about machine learning.
- What is the sampling frequency of the audio data?
- What is the unit of ordinate in Figure 5? Figure 6 doesn't even have coordinates.
- The authors mentioned “transfer learning” in line 351. However, it seems the work of this paper is irrelevant to “transfer learning”. So, it is necessary to point out the part related to “transfer learning”.
- In fact, the spectrogram images are set as the input of pre-trained 366 convolutional networks. So, it is necessary to point out the frequency resolution of the spectrogram images.
Reviewer 3 Report
The paper proposed CNN-based AI method to determine the severity level using acoustic data on gravel hitting the bottom of a car. The paper has application value and provides advanced guidance. The reviewers put forward the following minor suggestions.
1. Network structures are expected to be shown, including network details.
2. How sensitive is the CNN-based method?
3. Hundreds of training samples maybe not enough? How do the methods compared in this article perform on new data?
4. Fig.3 may too small and hard to read.
5. Table 2, It can be seen that the accuracy rates of different methods are relatively similar, which method has more advantages in practical application?
Round 2
Reviewer 1 Report
The authors addressed all the reviewer's comments with sufficient attention and modified the paper consistently with the suggestions provided. The new version of the paper has improved significantly both in the presentation that is now much more accessible even by a reader not expert in the sector, and in the contents that now appear much more incisive. The addition of a sufficient reference bibliography gave consistency to the authors' statements and the results they achieved.
Minor revision
Try to enrich the captions of the figures, the reader should be able to read the figure without the need to retrieve the information in the paper. Try to summarize the essential parts of the Figure and what you want to explain with it.
You need to be clearer in labelling data. This is the essential part of the job, the labeling. Explain what you mean by gravel or non-gravel.
169) Clarify the classes you have used in your work. In this table there are 4 classes but then you talk about a binary classification.
206) Table 3: Correct the car model name (Passa GTE 2018 ?). I also asked to indicate the height from the ground, for the study you have carried out it is crucial.
237-238) Explain better what you mean
286-290) Link this classification with the one shown in Table 1
385-386) Add more reference to study based on CNN, for example: “Sound event detection in underground parking garage”.
414-435) In order to make an effective comparison you should also enter information on the computational cost in order to understand if it is appropriate to use more expensive technologies from a computational point of view.
Reviewer 2 Report
My suggestion is accept in present form
Author Response
Dear Reviewer,
Thank you for recommending our article for publishing. Your comments have helped us improve our article significantly.
Best regards,
Nausheen.